# Prospective DNA Methylation Analysis of the CpG GABRA2 Receptor Subunit in Alcohol Dependence during Detoxification

**DOI:** 10.3390/medicina58111653

**Published:** 2022-11-15

**Authors:** Ulrich W. Preuss, Gabriele Koller, Peter Zill

**Affiliations:** 1Department of Psychiatry, Psychotherapy and Psychosomatics, Martin-Luther-University, Halle-Wittenberg, Julius-Kühn-Str. 7, 06112 Halle/Saale, Germany; 2RKH Hospital Psychiatry, Psychotherapy and Psychosomatic Medicine, Posilipostr. 4, 71640 Ludwigsburg, Germany; 3Department of Psychiatry and Psychotherapy, Ludwig-Maximilians-University Munich, Nussbaumstrasse 7, 80336 Munich, Germany; 4Division of Epigenetics, Ludwig-Maximilians-University Munich, Nussbaumstrasse 7, 80336 Munich, Germany

**Keywords:** alcohol dependence, alcohol intoxication, alcohol exposition, alcohol withdrawal, GABRA2, epigenetics, CpG methylation, neuroblastoma cell cultures, peripheral blood cells

## Abstract

*Background and Objectives:* Variants of GABRA2 have been repeatedly associated with alcohol dependence risk. However, no study investigated potential epigenetic alterations in the GABRA2 gene in alcohol-dependent (AD) subjects during alcohol withdrawal. We investigated DNA methylation pattern in the regulatory region of GABRA2 gene in peripheral leukocytes of AD patients and controls. Further, GABRA2 methylation patterns were analysed in neuroblastoma cells under ethanol exposure and withdrawal. *Materials and Methods:* In the present study, blood samples were obtained from 41 AD subjects on the day of inpatient admission, after the first and second week of inpatient treatment. The comparison group included 47 healthy controls. GABRA2 methylation of 4 CpG sites in the CpG island was compared to neuroblastoma cells which were exposed to 100 mM of ethanol for 2, 5 and 9 days, followed by a withdrawal interval of 4 days. *Results:* no significant differences in GABRA2 methylation patterns were found in AD subjects over time and vs. controls, after controlling for age. Further, no influence of withdrawal severity, alcohol consumption before admission and other alcohol dependence characteristics were found. *Conclusions:* The results indicate that GABRA2 methylation in AD individuals and in a cell model is unaffected by alcohol exposition and withdrawal. Influences of GABRA2 on characteristics of alcohol dependence may be exerted by mechanisms other than epigenetic alterations related to alcohol intoxication or withdrawal.

## 1. Introduction

Methylation of cytosine-phosphatidyl-guanine (CpG) sites within the promoter region of a specific gene are known to be an epigenetic mechanism that regulates gene expression [1]. Several findings suggest that ethanol-induced changes in DNA methylation may be an important factor in the regulation of gene expression that occurs during the complex processes of alcohol use disorder pathogenesis [2]. DNA methylation represses gene expression via the actions of a protein called methyl CpG-binding protein-2 (MeCP2), which selectively binds to methylated DNA, thereby blocking transcription.

Gamma-aminobutyric acid (GABA) is the main inhibitory brain neurotransmitter, and alcohol is thought to produce many of its effects by facilitating GABA receptor functioning [3,4]. GABA activity in the amygdala is enhanced by exposure to alcohol and then decreased during alcohol withdrawal [5]. GABAA receptors are large proteins embedded in the cell membranes of neurons. Each receptor consists of five protein molecules, or subunits, that assemble so that a channel is formed at the centre of the complex. Many different GABAA receptor subunits have been identified. These are sub-grouped into α, β, γ, δ, ε, π, θ subunits, of which the α4β1δ subtype seems to be more sensitive to ethanol than other receptor subtypes [6]. 

Each of these groups contains several different subunits (e.g., α2). Some of the subunit genes are located on chromosome 4 (including 4p13-q11: α2, α4, β1, γ1), in a region which has been repeatedly associated with alcohol dependence by linkage analysis [7] and GWAS [8]. 

These chromosome 4 clusters of genes, including the GABRA2 gene, are likely to be important in addiction and anxiety. They may be vulnerable to epigenetic effects [9] and may show a subjective response to alcohol and related phenotypes [10,11] but so far we have negative findings which do not support this hypothesis [12]. 

Chronic alcohol exposure regulates GABAA receptor function via both genomic and non-genomic mechanisms that interact to produce the complex adaptations in the brain [13]. It was suggested that the GABAA receptors are involved in alcohol dependence in both animals and humans. Alcohol exposure induces upregulation of GABA receptors containing α4, γ1 and most importantly α2 subunits [14], in particular α4βγ2 and α2β1γ1 GABAA receptor pentamers subtypes [15]. 

Withdrawal from alcohol and particularly repeated withdrawals markedly augment CNS excitability, which is a key characteristic of alcohol dependence. Substantial evidence indicates that withdrawal-related behavioural and neural adaptations are related to concomitant adaptations in the pharmacological properties of GABAA receptors. The upregulation of receptor subunits was reported to alleviate withdrawal symptoms when ethanol is absent [12]. Further, during withdrawal from alcohol, an increase in receptor density is expected. Indeed, a single-photon emission computed tomographic scan (SPECT) imaging study showed that, at one week of abstinence from alcohol, binding of 123I-iomazenil to GABAA receptors was higher throughout the brain, including the hippocampus and amygdala, but particularly in the frontal cortex, in alcoholic non-smokers compared with controls [16].

Regulatory mechanisms, which may modulate GABRA2 transcription on chromosome 4pq, include promoter proximal CpG island methylation. GABRA2 promoter proximal CpG island methylation may also change during prolonged times of alcohol intake and subsequent withdrawal. Increased promoter CpG island methylation is typically negatively correlated with gene expression and vice versa [17,18]. Thus, changes in CpG methylation may be responsible for altered GABRA2 gene expression during periods of intensive alcohol use and subsequent withdrawal. 

A relevant subgroup of patients has an early onset of their dependence before the age of 25 years. This early onset of alcohol use disorder corresponds to the so-called type II alcoholism subgroup according to the Cloninger’s classification [19]. Type II or “male-limited” alcoholism is suggested to be strongly heritable (estimated heritability of 90%), primarily transmitted from father to son, and showing moderate environmental influence [19]. Moreover, early onset of alcoholism often reflects greater severity, including a higher risk of recurrence.

We investigated promoter methylation of the GABRA2 receptor gene using two approaches. First, methylation of 4 individual CpG-sites in peripheral blood of alcohol-dependent patients (*n* = 47 males) vs. age- and gender-matched controls (*n* = 41 males) at admission to detox, during and after alcohol withdrawal was analysed. Further, characteristics of alcohol dependence including alcohol intake before admission, number of DSM-5 criteria, age of onset of alcohol dependence, number of current withdrawal symptoms and early vs. late onset alcohol dependence were compared within the patient group. 

Secondly, the same methylation pattern in neuroblastoma cell cultures were employed and were exposed to ethanol and underwent subsequent alcohol withdrawal. 

We hypothesize that the GABRA2 CpG methylation of alcohol-dependent individuals would change significantly during withdrawal. Similarly, neuroblastoma cells are expected to undergo significant changes in GABRA2 CpG methylation during ethanol exposition over 9 days and subsequent alcohol withdrawal. 

## 2. Materials and Methods 

### 2.1. Sample

#### Alcohol-Dependent Individuals

Treatment-seeking inpatient alcohol-dependent subjects were recruited from an addiction treatment ward of the Ludwig-Maximilian’s-University of Munich, Germany, during September 2019 and March 2020. 

All alcohol-dependent individuals were admitted via an outpatient motivational group. Subjects were excluded if they had any current and acute Axis I disorder, other than alcohol- and nicotine dependence. The university hospital offers an inpatient ‘qualified detoxification’ treatment program, which includes not only somatic detoxification, but also individual and group psychotherapy, counselling for social and financial problems as well as somatic medical care for at least 15 days. This approach is in line with the updated German S3-guidelines for inpatient treatment of alcohol-dependent individuals [20]. 

Blood samples were taken on the day of admission in the hospital, after 7 and 15 days of inpatient treatment.

All patients were unrelated, of German descent, 18 years and older, and met DSM-5 criteria for alcohol dependence, assessed with a structured clinical interview SSAGA (Semi-Structured Assessment on the genetics in alcoholism). The diagnostic assessment was performed without knowledge of methylation status. Demographic variables and clinical characteristics are presented in Table 1.

### 2.2. Control Group

The control persons were recruited from the general population at different locations (e.g., libraries, road construction sites and department stores). These individuals represent a wide range of social classes from university graduates to unskilled workers. All controls were comprehensively assessed to exclude medical and mental diseases including Axis I/II disorders, such as schizophrenia, depression, personality disorders or any alcohol- and substance use disorders by a short, structured interview with an experienced psychiatrist, as well as with the personality questionnaires (MMPI, NEO-FFI, TCI) and a routine laboratory screening. Further, individuals were excluded if they had any current Axis I disorder, any substance use disorder other than nicotine or any physical illness. Amount of alcohol intake (g/d) during the last month and smoking status were assessed. None of the subjects were related, all were of German descent and did not have first-degree relatives with any mental disorders (for details see [20]).

### 2.3. Neuronal Cell Cultures 

Human SH-SY5Y neuroblastoma cells (Thermo Fisher Scientific, Waltham, MA, USA) were grown in Dulbecco’s modified Eagle’s medium (DMEM, Biochrom, Berlin, Germany) with 10% foetal bovine serum (Biochrome, Berlin, Germany). Cells were seeded in 25 mm^2^ flasks at a density of 2 × 10^5^ cells per flask. Cells were kept in 5% CO_2_, 95% atmosphere with high humidity at 37 °C. After 4-day incubation, EtOH (100 mM) was added every 48 h during medium replenishment, if necessary.

Cells were harvested using Trypsin/EDTA (0.25%/0.02% *v*/*v*) (Biochrom, Berlin, Germany) and washed twice with PBS before DNA isolation.

To investigate a possible dynamic effect of ethanol on DNA-methylation, we measured the changes in DNA methylation of 4 CpG sites in the CpG island region of the GABRA2 gene in SH-SY5Y neuroblastoma cells after incubation with ethanol at different time points. We treated cell lines for 2, 5, and 9 days with ethanol (100 mM) and additionally screened for a “withdrawal” effect by incubation of one cell line with ethanol for 9 days, followed by an ethanol-free interval of 4 days.

In order to be able to achieve a clear alcohol effect on the DNA methylation, we chose a relatively high concentration of 100 mM ethanol. This approach was supported by numerous publications in which neuroblastoma cells were incubated with ethanol concentrations between 50 mM and 200 mM. As a result, numerous cellular reactions could be observed, but never cell death [21,22,23,24]. 

### 2.4. Promoter CpG Island Analysis

We selected a 10020 bp region (−6334/+3686, related to the ATG translation start site of transcript variant 1), containing the putative promoter and exon 1 of transcript variant 1 of the GABRA2 gene (NT_006238.12: nt 13546001 to 13556021). 

CpG content analysis was performed applying the CpG Island Searcher (http://www.Uscnorris.com/cpgislands2/cpg.aspx (accessed on 10 November 2010)). The CpG islands were defined as a region with at least 500 bp, with a GC percentage that is greater than 60% and with an observed/expected CpG ratio that is greater than 65%. 

The CpG island analysis revealed a 1219 bp CpG island (−2095 to −876, related to the ATG translation start site of transcript variant 1). Within some older bisulphite sequencing approaches by capillary electrophoresis of our group, we found intensive methylation from nucleotide −1843 to −1533. In the other sequenced parts of the CpG island, there was no methylation or, rarely, negligible methylation. Thus, we focused on the methylated region from nucleotide −1843 to −1533. The (−1843/−1533) region contains 30 CpG sites; 4 of them could be analysed in the present study by pyrosequencing with commercially available CpG assays. These 4 CpG sites were chosen for their match to available commercial CpG assays. For the other sites, there were no matching assays available.

### 2.5. Sequencing of Genomic DNA

The pyrosequencing reaction was carried out on the Pyromark Q48 Autoprep machine (Qiagen, Hilden, Germany) using a commercially available GABRA2 Pyromark CpG Assay (Hs_GABRA2_01_PM Pyromark CpG assay, Catalog No. PM0001901, Qiagen, Hilden, Germany) according to the manufacturer’s instructions. CpG methylation for four CpGs were assessed (positions −1817, −1813, −1805 and −1803 related to the ATG translation start site of transcript variant 1 (Figure 1).

The output data (obtained from Pyromark Q24 sequencing reactions), representing percent methylated cytosines over each of these four CpGs, was averaged for each sample analyzed.

### 2.6. Statistical Analysis

The sample size calculation was performed using the GPower 3.1.9.7 software (http://www.gpower.hhu.de/(accessed on 1 September 2022)) and the psychometric website (https://www.psychometrica.de/effect_size.html#nonparametric, (accessed on 1 September 2022)). It was based on the effect size reported from a previous epigenetic gene study on GABRA2 promoter methylation in schizophrenic patients vs. controls [25]. The calculation yielded an actual power of 0.95 and a sample size of *n* = 46 in patients and in controls (for both groups *n* = 92) with an effect size of 0.3761.

Statistical analysis was performed using SPSS (v25, IBM, Armonk, NY, USA).

All continuous variables were tested for deviation from normal distribution as calculated by Kolmogorov-Smirnov non-parametrical tests. Since all variables yielded significant deviation from normal distribution (Kolmogorov-Smirnov tests), non-parametrical tests (Mann-Whitney-U Test) were employed for subsequent analyses. The rate of smokers vs. non-smokers was compared using Chi-square statistics. 

To evaluate the influence of several important alcohol-dependence characteristics on CpG site methylation, a median split of the variables was conducted (alcohol intake before admission, number of DSM-5 criteria, age of onset of alcohol dependence, number of current withdrawal symptoms, and the subgroups early vs. late onset of alcohol dependence). 

Alongside descriptive statistics, age and early age vs. late age onset of alcohol dependence were compared between AD subjects and controls, and within AD individuals using Mann-Whitney-U statistics. 

Relationships between age and CpG island methylation level were computed using Spearman’s correlation coefficients (rho) while influence of age (median split) was tested using Mann-Whitney-U statistics. 

We used ANOVA repeated measures to examine whether 4 GABRA2 promoter-proximal CpG island differed between alcohol-dependent individuals and controls or altered across the 3 measurements during withdrawal in AD individuals and in neuroblastoma cells. Statistical significance was defined as *p* < 0.0125, since correction for multiple testing (Bonferroni) was applied and four CpG sites were included into the analyses. 

## 3. Results 

### 3.1. Sample 

We investigated global DNA methylation, as well as DNA methylation pattern of 4 CpG sites in the 5′-region of the GABRA2 gene in peripheral leukocyte DNA from 47 alcohol-dependent patients and 41 healthy controls. 

A significant difference between alcohol-dependent individuals and healthy controls was found for age (Mann-Whitney U-value 679, df = 87, *p* = 0.007). Age was used as a co-factor in subsequent analyses comparing alcohol-dependent individuals and controls. Within the alcohol-dependent patients and controls, one significant correlation between CpG methylation sites and age were found. For subsequent analyses within the alcohol-dependent group, age was used as a co-factor. 

Correlation between CpG site methylation in alcohol-dependent individuals and controls are presented in Table 2.

Smoking status (smokers vs. non-smokers) did not influence any CpG site methylation in both patients and controls, s. Table 3). For subsequent analyses, these variables were not considered as potential confounders or covariates.

### 3.2. Comparison of GABRA2 Methylation between AD Patients (Admission) and Controls

No significant differences between AD patients (ad admission) and controls were detected regarding CpG site 1–4 methylation, using age as a covariate (F-values 0.16, 0.41, 1.02, 4.92 df = 2, *p* = 0.68, 0.60, 0.42 and 0.21, respectively, see Figure 2).

### 3.3. Comparison of GABRA2 CpG Methylation Sites over Time in AD Subjects 

Using ANOVA repeated measures, no significant differences over the three measurements were detected for CpG site 1 (F-value: 0.06, df 1, *p* = 0.80), CpG site 2 (F-value: 0.56, df 1, *p* = 0.46), CpG site 3 (F-value: 1.73, df 1, *p* = 0.20) and CpG site 4 (F-value: 0.37, df 1, *p* = 0.55, see Figure 2). 

### 3.4. Influence of Alcohol Dependence Characteristics on GABRA2 CpG Methylation Sites over Time in AD Subjects

Using ANOVA repeated measurements for CpG sites 1–4 over time, no significant differences between groups were found for all alcohol dependence characteristics. Results are presented in Table 4. In Table 5, results of CpG methylation sites comparing early vs. late onset alcohol-dependence individuals are presented. No significant difference was found.

### 3.5. Neuroblastoma Cell Cultures

No methylation was observed in the four GABRA2 CpG sites in the incubated neuroblastoma cells. Further, there were no statistically significant differences in the methylation frequencies of the analysed four CpG sites between mock and treated cell lines over exposition period. Finally, we could not observe any effect of a nine-day EtOH incubation followed by a four-day “withdrawal” condition on DNA methylation compared to an untreated control sample. 

## 4. Discussion 

The aim of this article is two-fold. First, we measured DNA methylation of 4 CpG sites in the GABRA2 promoter region in a group of alcohol-dependent individuals during alcohol withdrawal at three time points over a period of 14 days. First blood sample was taken at patient’s admission to the addiction treatment ward, the second sample after the acute withdrawal and the third sample at the end of treatment after 2 weeks. Between AD patients and controls, and within alcohol-dependent patients over time, when age was used as a co-factor, no significant differences in GABRA2 promoter CpG site 1–4 methylation could be detected. Previous research indicated that older age is related to higher CpG methylation in general e.g., [26]. At baseline, GABRA2 CpG site 4 was positively correlated with age. However, subsequent comparison of this site over time did not reveal any significant difference between patients and controls, across measures. 

In a parallel design, neuroblastoma cells were exposed to alcohol (100 mM) over a time of 9 days. Subsequently, alcohol was withdrawn. We could not find any methylation at the 4 CpG sites, thus, no significant alterations in GABRA2 promoter CpG site 1–4 methylation could be detected in both the period of alcohol exposition and subsequent withdrawal. In summary, the hypothesis of GABRA2 methylation changes in AD subjects could not be confirmed. 

We are aware that a limiting factor of the present study is the use of the high ethanol concentration of 100 mM in the cell culture experiments, compared to normal physiological concentrations in humans. The blood alcohol content is normally given in grams of ethanol per 1000 g of blood. A total of 1 g ethanol/ 1000g blood corresponds to a concentration of 21.7 mmol/L [27]. Therefore, 100 mM corresponds to an alcohol concentration of 4.6 g per 1000 g of blood. This is higher than the average blood alcohol concentration in alcohol-dependent subjects. Moreover, it has been reported that in vitro alcohol treatment using concentrations above 100 mmol/L has a direct cytotoxic effect of alcohol on cells [28]. On the other hand, several recent studies could not confirm this observation [29,30,31]. To investigate whether 100 mM ethanol is toxic to neuroblastoma cells or not, viability of cells should have been tested. Unfortunately, this experiment was not conducted for the present analyses and should be considered in future studies.

From the present results, it seems unlikely that ethanol influences the DNA-methylation pattern in neuroblastoma cells, but considering the limitations, we cannot finally exclude an ethanol effect and further studies are needed. 

Other characteristics of alcohol use disorders, such as average alcohol intake during the last month, withdrawal severity or duration of alcohol dependence, had no influence on GABRA2 methylation patterns. Further, no difference was detected comparing early vs. late onset alcohol-dependent individuals. To our knowledge, this is the first study to investigate GABRA2 methylation pattern during alcohol withdrawal in AD subjects. The negative results comparing alcohol-dependent individuals and controls, as well as alcohol-related phenotypes and smoking within patients, indicate that methylation patterns in GABRA2 are quite stable, even comparing early (more genetically influenced) vs. late onset patients. Methylation is also not significantly influenced by alcohol consumption, withdrawal symptoms or other relevant phenotypes such as smoking status although smoking, alcohol intake and age are significantly different between patients and controls. 

Previous research related GABRA2 genetic variants, but not methylation patterns to endophenotypes of alcohol response and alcohol use disorder risk. An initial study reported linkage of GABRA2 gene with alcohol-related phenotypes such as β-EEG power and excess fast EEG activity [10]. Further, GABRA2 gene variants were also associated with impulsive behaviours measured in an incentive delay task, which increases the risk of development and relapse of alcohol use disorders [32]. 

Moreover, subjective responses to alcohol have been associated with polymorphisms of GABRA2. Three SNPs (rs279858, rs573400, rs279871) were related to the rewarding effects of alcohol administration. These results confirm the role of GABRA2 variants in several biobehavioural mechanisms which may influence the risks of development and maintenance of alcohol use disorders [11]. 

However, results of human genetic association studies remain contradictory. Previous studies reported no evidence of an association between GABRA2 polymorphisms and alcohol dependence [33,34]. Even protection from alcohol dependence has been attributed to GABRA1 and GABRA2 polymorphisms in a small study from India [35], probably due to ethnic stratification effects [36].

Recent studies investigated whether GABRA2 genetic variants influence gene expression. Using induced pluripotent stem cell (iPSC)-derived neural cultures, it was reported that the AUD-associated GABRA2 synonymous polymorphism rs279858 was associated with altered gene expression of the chr4p12 GABAA subunit gene cluster. The absence of a parallel effect in the post-mortem human adult brain samples suggests that alcohol dependence-associated genotype effects on GABAA expression, although not present in mature cortex, could have effects on regulation of the chromosome 4p12 GABAA cluster during neural development [37]. In a subsequent study from the same research group, it was noted that the GABRA2 CpG promoter island undergoes random stochastic methylation during reprogramming of iPSC-derived neural cultures and that methylation is associated with decreased GABRA2 gene expression [18]. In comparison to our neuroblastoma cell cultures of the present study, an intermediate level of methylation was detected in their cell lines.

Taken together, there is evidence that GABRA2 genetic polymorphisms and functional consequences contribute to the risk of alcohol dependence and related phenotypes and have detectable but minor effects on DSM-defined AUD [11,38]. Further, changes in GABA receptor composition may occur during prolonged times of alcohol intake and repeated withdrawal [14]. 

The major finding of this study is that there is no change in methylation patterns of four GABRA2 CpG sites in the CpG island during intoxication and withdrawal in AD individuals admitted for (qualified) alcohol withdrawal treatment. No significant changes in the same GABRA2 CpG site methylation patterns were found in neuroblastoma cell lines which were exposed to alcohol over a period of 9 days and subsequent ethanol withdrawal. 

Regarding alcohol-dependent individuals with more than 10 years of dependence, GABRA2 methylation changes may have occurred in the past, but they may be persistent, as reported from a rat model, which reported persistence after 30 or more doses, lasting for at least 120 days [14]. 

Compared to previous studies, we used a different cell model (neuroblastoma cells) [15,34] instead of stem-cells. These neuroblastoma cells, such as fibroblasts, may have low levels of methylation at the GABRA2 site. This might also explain the differing results in our compared to studies on stem cells. 

Therefore, if there are alterations of GABRA2 function in alcohol-dependent individuals, changes in GABA receptor composition and GABRA2 subunits may be exerted by several SNPs in the GABRA2 gene and other related epigenetic mechanisms such as histone modifications and RNA interferences or post-translational modifications of the GABRA2 protein [39]. These alterations were not exerted via methylation pattern changes due to alcohol intoxication or during alcohol withdrawal.

Of course, the present study has several limitations. First, the number of subjects included into the AD sample and controls, despite antedated sample size analysis, is rather small and may not represent all alcohol-dependent individuals with their biological and methylation pattern or their clinical characteristics. This may be one reason for the negative findings. Further, since most alcohol-dependent individuals in qualified withdrawal treatment are males (up to three-quarters of patients), females were excluded from the study. 

Regarding the cell lines, half of the study was based on in vitro experiments in neuroblastoma cell lines, so it is possible that the experimental conditions did not accurately reflect those of an in vivo brain. Third, only four out of more than twenty GABRA2 CpG sites were investigated regarding their methylation pattern changes over time, due to methodological reasons. Furthermore, we could not detect any effect of ethanol on the GABRA2 methylation pattern in neuroblastoma cells. Ethanol was added every 48 h, which means that the ethanol concentrations were not constant, due to high rate of alcohol evaporation. It is possible that these cells in culture experienced many small ethanol withdrawals. Following studies should have the cells incubated in alcohol-saturated atmosphere and change the medium with alcohol every day. Moreover, we did not elucidate the functional significance of DNA methylation changes in this study. Comprehensive gene expression analysis will help elucidate the consequences of changes in DNA methylation. Future research may investigate whether these 4 CpG sites are representative of all GABRA2 promoter methylation sites and whether these sites show causal relations.

## Figures and Tables

**Figure 1 medicina-58-01653-f001:**
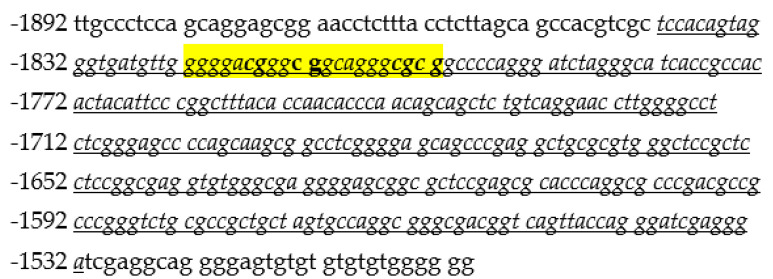
Sequence of the (−1843/−1533) region (italic, underlined) including the 4 investigated CpG sites (bold).

**Figure 2 medicina-58-01653-f002:**
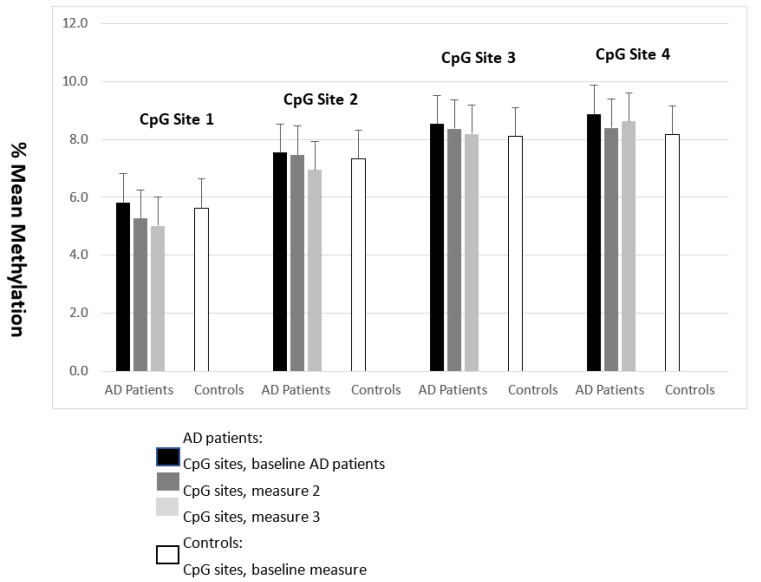
GABRA2 CpG Site Methylation in Alcohol-Dependent Individuals vs. Control Subjects Across Measurements (Mean + SD).

**Table 1 medicina-58-01653-t001:** Demographic Variables and Clinical Characteristics of Alcohol- Dependent Individuals and Healthy Controls.

	AD Patients	Controls	*p*
Gender (males/females)	47/0	44/0	
Age (years, mean ± SD)	44.83 ± 10.90	37.98 ± 12.29	0.007
Age of onset (years, mean ± SD)	23.48 ± 9.89		
Mean daily alcohol intake before admission (g/d, mean ± SD)	190.43 ± 150.90	4.54 ± 4.17	<0.001
Maximal daily alcohol intake ever (g/d, mean ± SD)	395.03 ± 200.93		
Duration of dependence (years, mean ± SD)	21.92 ± 13.30		
Current withdrawal symptoms (#, mean ± SD)	3.09 ± 2.36		
DSM-5 criteria (#, mean ± SD)	5.91 ± 2.66		
Total Oxazepam medication during withdrawal (mg, mean ± SD)	130.45 ± 296.79		
Tobacco users *n*	33 (75%)	3 (7%)	<0.001

Abbreviations are: SD: standard deviation of mean; #, *n*: Number, g/d: gram per day, mg: milligram.

**Table 2 medicina-58-01653-t002:** Correlation between Age and Baseline GABRA2 CpG Methylation in Alcohol-dependent Individuals and Controls (Spearman’s rho).

	AD Patients	*p*	Controls	*p*
CpG Site 1 baseline	−0.061	0.687	0.205	0.182
CpG Site 2 baseline	0.159	0.290	0.032	0.838
CpG Site 3 baseline	0.051	0.735	0.272	0.074
CpG Site 4 baseline	0.102	0.499	0.415	0.005 **

**: *p* > 0.01

**Table 3 medicina-58-01653-t003:** Smoking Status and GABRA2 CpG Site Methylation in Alcohol-Dependent Patients and Controls.

	AD Patients Non-Smoking	AD Patients Smoking	M-W-U Value, Sign. (2-Sided)	Controls Non-Smoking	Controls Smoking	M-W-U Value, Sign. (2-Sided)
Number of individuals	14	33		41	3	
GABRA2 CpG site 1, baseline	5.31 ± 0.88	4.65 ± 0.85	51.0; 0.18	5.63 ± 1.29	5.63 ± 0.50	55.5; 0.79
GABRA2 CpG site 1, measure 2	5.74 ± 1.61	4.91 ± 1.18	53.0; 0.53			
GABRA2 CpG site 1, measure 3	4.92 ± 0.85	4.67 ± 1.43	50.0; 0.26			
GABRA2 CpG site 2, baseline	7.22 ± 0.90	6.56 ± 1.93	52.0; 0.23	7.36 ± 1.25	6.66 ± 0.94	38.0; 0.29
GABRA2 CpG site 2, measure 2	8.18 ± 2.04	6.84 ± 1.72	48.0; 0.15			
GABRA2 CpG site 2, measure 3	6.91 ± 0.74	6.64 ± 2.08	44.0; 0.64			
GABRA2 CpG site 3, baseline	7.98 ± 1.67	7.31 ± 1.48	53.5; 0.53	8.06 ± 1.17	8.51 ± 0.71	46.0; 0.50
GABRA2 CpG site 3, measure 2	9.24 ± 0.91	7.73 ± 1.48	49.0; 0.20			
GABRA2 CpG site 3, measure 3	8.01 ± 1.59	7.76 ± 1.53	56.0; 0.88			
GABRA2 CpG site 4, baseline	8.72 ± 1.66	7.78 ± 1.44	53.0; 0.53	8.15 ± 1.25	8.14 ± 1.24	58.5; 0.89
GABRA2 CpG site 4, measure 2	8.89 ± 1.15	8.30 ± 1.98	53.0; 0.53			
GABRA2 CpG site 4, measure 3	8.68 ± 1.35	7.80 ± 1.77	51.0; 0.18			

AD: Alcohol dependence; M-W-U: Mann-Whitney-U Test, sign.: Statistical significance *p*.

**Table 4 medicina-58-01653-t004:** Influence of Important Alcohol-Dependence Characteristics on GABRA2 CpG Methylation in Alcohol-Dependent Individuals (Median split: low vs. high, age as cofactor).

	AD Patients Alcohol Intake before Admission	ANOVA F-Value, Sign.	AD Patients Number of DSM-5 Criteria	ANOVA F-Value, Sign.	AD Patients Age of Onset	ANOVA F-Value, Sign.	AD Patients Number of Withdrawal Symptoms	ANOVA F-Value, Sign.
GABRA2 CpG site 1, baseline	4.74 ± 0.69 5.02 ± 1.15	0.559 0.590	5.05 ± 0.92 5.07 ± 1.70	0.351 0.711	5.10 ± 1.35 5.07 ± 0.98	0.493 0.622	4.87 ± 0.74 5.30 ± 1.57	0.780 0.482
GABRA2 CpG site 1, measure 2	5.08 ± 1.51 5.12 ± 0.65		5.41 ± 1.45 5.10 ± 1.42		5.02 ± 1.29 5.83 ± 1.58		5.81 ± 1.61 5.03 ± 1.25	
GABRA2 CpG site 1, measure 3	4.98 ± 0.87 4.63 ± 056		5.22 ± 0.93 4.67 ± 1.69		5.05 ± 1.51 4.85 ± 0.68		5.45 ± 1.08 4.72 ± 1.43	
GABRA2 CpG site 2, baseline	6.62 ± 0.69 6.92 ± 1.31	0.622 0.558	7.13 ± 1.57 6.57 ± 1.93	0.199 0.822	7.07 ± 1.88 6.92 ± 1.22	0.057 0.944	6.98 ± 1.75 6.98 ± 1.82	0.001 0.999
GABRA2 CpG site 2, measure 2	7.66 ± 1.95 6.95 ± 1.09		7.69 ± 1.94 7.04 ± 1.74		7.31 ± 1.57 7.99 ± 2.37		7.82 ± 2.43 7.34 ± 1.39	
GABRA2 CpG site 2, measure 3	6.95 ± 0.74 6.26 ± 0.92		7.33 ± 1.47 6.32 ± 1.77		6.92 ± 1.86 7.00 ± 1.13		7.52 ± 1.88 6.56 ± 1.49	
GABRA2 CpG site 3, baseline	7.37 ± 1.25 7.72 ± 2.08	0.218 0.809	7.97 ± 1.37 7.44 ± 2.23	0.206 0.817	7.50 ± 1.91 8.09 ± 1.55	0.179 0.838	7.46 ± 0.94 8.10 ± 2.20	0.224 0.834
GABRA2 CpG site 3, measure 2	8.16 ± 1.67 8.28 ± 1.36		8.67 ± 1.38 7.83 ± 1.87		7.93 ± 1.66 8.84 ± 1.78		8.58 ± 1.48 8.39 ± 1.71	
GABRA2 CpG site 3, measure 3	8.08 ± 1.53 7.86 ± 1.64		8.62 ± 1.25 7.44 ± 1.85		7.79 ± 1.80 8.56 ± 1.38		8.54 ± 1.33 8.03 ± 2.21	
GABRA2 CpG site 4, baseline	7.63 ± 1.01 8.68 ± 1.97	1.763 0.226	8.63 ± 1.58 8.06 ± 2.26	0.431 0.660	8.32 ± 2.20 8.20 ± 1.34	0.091 0913	8.02 ± 1.27 8.87 ± 2.21	0.490 0.625
GABRA2 CpG site 4, measure 2	8.22 ± 2.00 7.82 ± 1.54		8.50 ± 1.62 8.41 ± 2.43		8.40 ± 1.85 8.44 ± 2.39		8.27 ± 1.99 8.73 ± 2.04	
GABRA2 CpG site 4, measure 3	7.98 ± 1.46 8.94 ± 1.78		9.24 ± 1.53 7.95 ± 2.23		8.11 ± 2.20 8.97 ± 1.24		8.98 ± 1.32 8.85 ± 2.20	

Abbreviations: AD: Alcohol Dependence; sign.: statistical significance.

**Table 5 medicina-58-01653-t005:** Early vs. Late onset patients and GABRA2 CpG Site Methylation in Alcohol-Dependent Patients and Controls.

	AD Patients Early Onset (before Age 25)	AD Patients Late Onset (after Age 25)	M-W-U Value; Sign. (2-Sided)
Number of individuals	32	15	
GABRA2 CpG site 1, baseline	5.37 ± 1.19	6.00 ± 1.60	51.0; 0.495
GABRA2 CpG site 1, measure 2	5.34 ± 1.60	5.17 ± 0.86	50.0; 0.851
GABRA2 CpG site 1, measure 3	5.04 ± 1.43	4.84 ± 0.79	52.0; 0.234
GABRA2 CpG site 2, baseline	7.29 ± 1.48	7.53 ± 1.71	47.5; 0.754
GABRA2 CpG site 2, measure 2	7.70 ± 1.97	7.08 ± 1.42	49.5; 0.753
GABRA2 CpG site 2, measure 3	7.00 ± 1.79	6.79 ± 1.19	52.0; 0.234
GABRA2 CpG site 3, baseline	7.83 ± 1.53	8.57 ± 2.14	51.5; 0.495
GABRA2 CpG site 3, measure 2	8.13 ± 1.71	8.50 ± 1.87	49.0; 0.200
GABRA2 CpG site 3, measure 3	7.75 ± 1.72	8.88 ± 1.36	43.0; 0.188
GABRA2 CpG site 4, baseline	8.22 ± 1.67	8.57 ± 1.65	55.0; 0.530
GABRA2 CpG site 4, measure 2	8.56 ± 1.83	8.02 ± 2.53	48.0; 0.661
GABRA2 CpG site 4, measure 3	8.15 ± 2.10	9.07 ± 1.41	55.0, 0.180

AD: Alcohol dependence; M-W-U: Mann-Whitney-U Test, sign.: Statistical significance *p*.

## Data Availability

All data generated or analysed during this study are included in this article. Further enquiries can be directed to the corresponding author.

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
