# Peer review of "Prospective DNA Methylation Analysis of the CpG GABRA2 Receptor Subunit in Alcohol Dependence during Detoxification"

_medicina, 2022, doi:10.3390/medicina58111653_

Round 1

Reviewer 1 Report

The study investigated DNA methylation pattern in the regulatory region of the GABRA2 gene in peripheral leukocytes of AD patients and controls. Further, GABRA2 methylation patterns were analysed in neuroblastoma cells under ethanol exposure and withdrawal. The results are potentially interesting; however, there are many issues that need to be addressed.

General

1)      As already acknowledged by authors, despite favourite Gpower analysis, the study enrolled relatively low number of subjects.

2)      The mean age of onset of alcohol abuse is low (23,48 ± 9.89). It is obvious that many alcohol-dependent patients in this study had early onset of alcohol abuse (occurring before 25 years of age) and they are all males, which may correspond to the type II alcoholism subgroup according to the Cloninger's classification (Cloninger et al., 1988). Type II or “male-limited” alcoholism is suggested to be strongly heritable (estimated heritability of 90%), transmitted primarily from father to son, and showing moderate environmental influence (Cloninger et al., 1988). Moreover, early onset of alcoholism often reflects greater severity, including a higher risk for recurrence. Therefore, it might be important to separately evaluate/compare GABRA2 methylation in alcohol-dependent subjects with early and late onset of alcohol abuse.

3)      The manuscript is very poor written since the authors wrote many contradictory statements in the text (i.e. DSM-IV or DSM 5, 200 mM or 100 mM ethanol, no changes in the age but p=0.006; data of 3 or 4 CpG islands analysed, etc.).

4)      The results are very scarcely presented and they should be presented in tables and figures.

5)      In experiments with neuroblastoma cells the authors have used very high concentrations of ethanol (100 or 200 mM). These concentrations were repeatedly proved as very toxic to cells by inducing apoptotic cell death. Why have authors used so high concentrations of ethanol for a relatively long period-9 days? To what ethanol concentrations in human blood/brain do these concentrations correspond?

Introduction

1)      The authors wrote: „Many different GABA-A receptor subunits have been identified. These are sub-grouped into α, β and γ subunits.“ Although in mammalian tissue, the most common receptor subtype contains α1, β2 and γ2 subunits, there are also δ, an ε , a π , and a θ  subunits. Moreover, α4β1δ subtype seems to be more sensitive to ethanol than other receptor subtypes tested to date. Therefore, GABA-A receptors other than α, β and γ subunits should be mentioned in the Introduction.

2)      The authors wrote: „Upregulation of receptor subunits were reported to alleviate withdrawal symptoms when ethanol is absent [12].“ The authors should specify which GABA-A receptors are upregulated (α4 and most importantly α2 subunits).

3)      The authors wrote that neuroblastoma cell cultures were exposed to several dosages of alcohol and underwent subsequent alcohol withdrawal. However, in materials and method authors first mention 200 mM and later 100 mM ethanol. Which is true? Please correct.

Materials and methods

1)      The authors wrote that the university hospital offers an inpatient ‘qualified detoxification’ treatment program. Does this program also includes pharmacotherapy? Have alcohol-dependent patients received some medications during detoxification, such as benzodiazepines, anticonvulsants, baclofen, supplements or other drugs? This should be clearly stated in 2.1. Sample section. In Table 1 authors wrote Oxazepam medication during withdrawal. Is this only medication administered to alcohol-dependent patients? Have all patients received oxazepam?

2)      The authors wrote that all patients were unrelated, of German descent, older than 18 years and met ICD-10 and DSM-IV criteria for alcohol dependence. Why have authors used DSM-IV and not DSM-V criteria for diagnosis of alcohol dependence? I understand that ICD-11 was adopted very recently (1st January 2022); however DSM-V has been introduced in 2013. Moreover, in Table 1 the authors wrote DSM-5 criteria (#, mean ± stddev)? What is true?

3)      Why there are no female subjects enrolled? This is also a disadvantage of the study, which should be acknowledged.

4)      In what time-period (from year- to year?) the samples from AD patients and controls were collected?

5)      Are authors sure that AD patients didn't drink any alcohol during detoxification? Have they determined the blood alcohol concentration in AD patients on regular basis during study?

6)      How were the withdrawal symptoms assessed? It seems that AD patients have low number of withdrawal symptoms (0.54 ± 1.32), which is strange considering the duration of dependence (21.92 ± 13.30 years). Please discuss these findings.

7)      In Table 1, the authors should also add the smoking status of both, AD patients and control subjects. The authors wrote that for control subject the amount of alcohol intake (g/d) during the last month were also assessed. These data should be also added in Table 1.

8)      The authors first wrote “After 4-day incubation, EtOH (200 mM) was added every 48h during medium replenishment, if necessary”. Later the authors wrote “We treated cell lines for two, five and nine days with ethanol (100 mM) and additionally screened for a “withdrawal” effect by incubation of one probe with ethanol for 9 days, followed by an ethanol-free interval of 4 days.” So what concentration was used 200mM or 100 mM?

9)      Authors have used very high concentrations of ethanol (100 or 200 mM). These concentrations were repeatedly proved as very toxic to cells by inducing apoptotic cell death. The authors should show the viability of cells during two, five and nine days of ethanol treatment, and 4 day withdrawal.

10)   If medium with ethanol was added every 48h it means that ethanol concentrations were not constant, due to high alcohol evaporation. It is possible that these cells in culture experienced many small ethanol withdrawals. The authors should have incubated the cells in alcohol-saturated atmosphere and changed the medium with alcohol every day. This is also a disadvantage of the study.

11)   The authors wrote: “The (-1843/-1533) region contains 30 CpG sites; 4 of them could be analysed in the present study by pyrosequencing and commercially available CpG assays.” Please explain why the authors have chosen particularly these 4 CpG islands.

12)   The authors wrote: „The output data (obtained from Pyromark Q24 sequencing reactions), representing percent methylated cytosines over each of these three CpGs, was averaged for each sample analyzed.“  3 or 4 CpG island were analyzed? Please check.

13)   Statistical analysis: The authors wrote. “Since all variables yielded significant deviation from normal distribution (Kolmogorov-Smirnov tests), non-parametrical tests (Mann-Whitney-U Test or Kruskal-Wallis analysis of variance) was employed for subsequent analyses.” However, age was compared between AD subjects and controls using Student’s t statistics. If there is not normal distribution in age the authors should use Mann-Whitney-U Test. Moreover, the authors should then use non-parametric Spearman’s test for relationships between age and CpG island methylation level, as well as Mann-Whitney-U Test for testing the influence and age.

14)   The authors wrote that they used two-way ANOVA and MANOVA repeated measures. However, the results only of ANOVA repeated measures were mentioned. Please check.

15)   The authors wrote: “Correction for multiple testing (Bonferroni) was applied when necessary”. However, there is no evidence of any correction for multiple testing. Please describe in what why the correction for multiple testing is applied and what p-value was considered significant after correction (since 4 CpG islands were analysed- p value should be set at least at 0,0125).

Results

·         The authors wrote. „No significant difference between alcohol-dependent individuals and healthy controls were found for age (t-value 2,83, df = 87, p = 0,006).“ However, AD patients were statistically significantly older then control subjects (p=0.006). Please check this.

·         Please present in Table the results regarding correlations between CpG methylation sites and age in alcohol-dependent patients and controls.

·         Please present in Table the results regarding the influence of smoking status (smokers vs. non-smokers) on CpG site methylation.

·         There is no data presenting the percent methylated cytosines over each of these 4 CpGs. Please present the results (Figure) regarding the GABRA2 methylation in AD patients (admission) and controls, as well as regarding comparison of GABRA2 CpG Methylation sites over time in AD subjects.

·         Please present all data regarding influence of various alcohol dependence characteristics on GABRA2 CpG Methylation sites over time in AD subjects.

·         All data should be presented in forms of Tables and Figures and with statistical tests used specified for each analysis.

·         Please present the % of Methylation at four CpG sites of the GABRA2 gene in all groups of neuroblastoma cells (control, ethanol treated, withdrawal) as Figure.

·         In addition, present the viability of neuroblastoma cells during ethanol treatment, since these high concentrations of ethanol are very toxic to cells.

Discussion

Please add other limitations of the study (no female subjects, not controlling for early and late onset of alcohol abuse, too high toxic ethanol concentrations used for neuroblastoma treatment, etc.) and discuss them.

Minor

Common abbreviation for standard deviation is SD

There are some typos and language errors throughout the text, which need to be corrected.

Author Response

Manuscript medicina-1940573

Responses to Review 1:

The study investigated DNA methylation pattern in the regulatory region of the GABRA2 gene in peripheral leukocytes of AD patients and controls. Further, GABRA2 methylation patterns were analysed in neuroblastoma cells under ethanol exposure and withdrawal. The results are potentially interesting; however, there are many issues that need to be addressed.

General

  • As already acknowledged by authors, despite favourite Gpower analysis, the study enrolled relatively low number of subjects.

Response: low number of subjects are mentioned in the limitation of the study paragraph at the end of the discussion section. This may be one of the reasons for the negative findings.

  • The mean age of onset of alcohol abuse is low (23,48 ± 9.89). It is obvious that many alcohol-dependent patients in this study had early onset of alcohol abuse (occurring before 25 years of age) and they are all males, which may correspond to the type II alcoholism subgroup according to the Cloninger's classification (Cloninger et al., 1988). Type II or “male-limited” alcoholism is suggested to be strongly heritable (estimated heritability of 90%), transmitted primarily from father to son, and showing moderate environmental influence (Cloninger et al., 1988). Moreover, early onset of alcoholism often reflects greater severity, including a higher risk for recurrence. Therefore, it might be important to separately evaluate/compare GABRA2 methylation in alcohol-dependent subjects with early and late onset of alcohol abuse.

Response: n = 32 (68%) individuals had an age of onset under 25 years while n = 14 individuals. Subgrouping the individuals in “early vs. late” onset patients, however, did not show any significant differences across GABRA2 methylation at all time points (Mann-Whitney test level of significance between p = 0.28 and p = 0.85).  

  • The manuscript is very poor written since the authors wrote many contradictory statements in the text (i.e. DSM-IV or DSM 5, 200 mM or 100 mM ethanol, no changes in the age but p=0.006; data of 3 or 4 CpG islands analysed, etc.).

Response: Alle patients are diagnosed with DSM-5 criteria, now mentioned in the methods section. Age is different between patients and controls. Four GABRA2 CpG islands were investigated.

We used 100 mM ethanol in all experiments

  • The results are very scarcely presented and they should be presented in tables and figures.

        Responses: three tables (2-4) and one figure (2) wer added to the paper

5)      In experiments with neuroblastoma cells the authors have used very high concentrations of ethanol (100 or 200 mM). These concentrations were repeatedly proved as very toxic to cells by inducing apoptotic cell death. Why have authors used so high concentrations of ethanol for a relatively long period-9 days? To what ethanol concentrations in human blood/brain do these concentrations correspond?

Response: We used 100 mM ethanol in all experiments. The concentration of 100 mM ethanol was chosen based on numerous other publications in which neuroblastoma cells were incubated with ethanol concentrations between 50 mM and 200 mM. As a result, numerous cellular reactions could be observed, but never cell death (Barann et al. 1995; Basavarajappa and Hungund 1999; Charness etal. 1993; Kelly et al. 1995). We do not know to what ethanol concentrations in human blood/brain these concentrations correspond.

 M Barann , K Ruppert, M Göthert, H Bönisch Increasing effect of ethanol on 5-HT3 receptor-mediated 14C-guanidinium influx in N1E-115 neuroblastoma cells. Naunyn Schmiedebergs Arch Pharmacol 1995 Aug;352(2):149-56.

B S Basavarajappa , B L Hungund Chronic ethanol increases the cannabinoid receptor agonist anandamide and its precursor N-arachidonoylphosphatidylethanolamine in SK-N-SH cells

J Neurochem. 1999 Feb;72(2):522-8.

M E Charness , G Hu, R H Edwards, L A Querimit Ethanol increases delta-opioid receptor gene expression in neuronal cell lines Mol Pharmacol. 1993 Dec;44(6):1119-27.

E Kelly , P K Harrison, R J Williams Effects of acute and chronic ethanol on cyclic AMP accumulation in NG108-15 cells: differential dependence of changes on extracellular adenosine

Br J Pharmacol. 1995 Apr;114(7):1433-41.

Introduction

  • The authors wrote: „Many different GABA-A receptor subunits have been identified. These are sub-grouped into α, β and γ subunits.“ Although in mammalian tissue, the most common receptor subtype contains α1, β2 and γ2 subunits, there are also δ, an ε , a π , and a θ Moreover, α4β1δ subtype seems to be more sensitive to ethanol than other receptor subtypes tested to date. Therefore, GABA-A receptors other than α, β and γ subunits should be mentioned in the Introduction.

Response: GABA-A receptor subtypes other than α, β and γ subunits are now mentioned in the introduction section, including alcohol-sensitivity of α4β1δ subtype.

  • The authors wrote: „Upregulation of receptor subunits were reported to alleviate withdrawal symptoms when ethanol is absent [12].“ The authors should specify which GABA-A receptors are upregulated (α4 and most importantly α2 subunits).

Response: the upregulation of α4 and α2  GABA-A subunits is now specified in the introduction section. In particular α4βγ2, subtypes, as well as increased levels of GABAAR α2 and γ1 subunits, along with increased α2β1γ1 GABAAR pentamers are mentioned (Lindenmeyer et al 2017)

  • The authors wrote that neuroblastoma cell cultures were exposed to several dosages of alcohol and underwent subsequent alcohol withdrawal. However, in materials and method authors first mention 200 mM and later 100 mM ethanol. Which is true? Please correct.

Response: We used 100 mM ethanol in all experiments

Materials and methods

  • The authors wrote that the university hospital offers an inpatient ‘qualified detoxification’ treatment program. Does this program also includes pharmacotherapy? Have alcohol-dependent patients received some medications during detoxification, such as benzodiazepines, anticonvulsants, baclofen, supplements or other drugs? This should be clearly stated in 2.1. Sample section. In Table 1 authors wrote Oxazepam medication during withdrawal. Is this only medication administered to alcohol-dependent patients? Have all patients received oxazepam?

Response: Yes, majority of patients (n = 43, 91.5%) received oxazepam for withdrawal (n = 91.5%). Table 1 has been corrected and mean total dose of oxazepam is presented. No significant relationship between total oxazepam dose and GABRA2 methylation changes was found. 

  • The authors wrote that all patients were unrelated, of German descent, older than 18 years and met ICD-10 and DSM-IV criteria for alcohol dependence. Why have authors used DSM-IV and not DSM-V criteria for diagnosis of alcohol dependence? I understand that ICD-11 was adopted very recently (1st January 2022); however DSM-V has been introduced in 2013. Moreover, in Table 1 the authors wrote DSM-5 criteria (#, mean ± stddev)? What is true?

Response: DSM-5 criteria are used to diagnose patients which were met by all included alcohol-dependent subjects.  The paragraph in the methods section has been modified accordingly.

3)      Why there are no female subjects enrolled? This is also a disadvantage of the study, which should be acknowledged.

      Response: males and females have different methylation patterns. Since a minority of alcohol-dependent individuals in treatment are females, they were excluded from the current study due to low sample size. This is noted in a paragraph “limitation of the study” at the end of discussion section. 

  • In what time-period (from year- to year?) the samples from AD patients and controls were collected?

Response: Patients were recruited from September 2019 to March 2020. Controls were included from a previous study (e.g. Preuss et al 2013). This information is added to the methods section.

  • Are authors sure that AD patients didn't drink any alcohol during detoxification? Have they determined the blood alcohol concentration in AD patients on regular basis during study?

Response: alcohol-dependent subjects are in inpatient treatment and are frequently screened for alcohol use, in particular when leaving the hospital e.g. for visits of self-help groups. No included subject was tested positive for alcohol use during treatment

6)      How were the withdrawal symptoms assessed? It seems that AD patients have low number of withdrawal symptoms (0.54 ± 1.32), which is strange considering the duration of dependence (21.92 ± 13.30 years). Please discuss these findings.

      Response: withdrawal symptoms were assessed using the SSAGA (semi-structured assessment of alcoholism) shortly after withdrawal was completed.  Accidentally, the controls were included into the initial analysis. After excluding the controls, the mean number of withdrawal symptoms are now 3.09 ± 2.36 symptoms. Still, there is no correlation to GABRA2-methylation.

7)      In Table 1, the authors should also add the smoking status of both, AD patients and control subjects. The authors wrote that for control subject the amount of alcohol intake (g/d) during the last month were also assessed. These data should be also added in Table 1.

      Response: daily alcohol intake and smoking status are included in table 1.

8)      The authors first wrote “After 4-day incubation, EtOH (200 mM) was added every 48h during medium replenishment, if necessary”. Later the authors wrote “We treated cell lines for two, five and nine days with ethanol (100 mM) and additionally screened for a “withdrawal” effect by incubation of one probe with ethanol for 9 days, followed by an ethanol-free interval of 4 days.” So what concentration was used 200mM or 100 mM?

Response: We used 100 mM ethanol in all experiments

9)      Authors have used very high concentrations of ethanol (100 or 200 mM). These concentrations were repeatedly proved as very toxic to cells by inducing apoptotic cell death. The authors should show the viability of cells during two, five and nine days of ethanol treatment, and 4 day withdrawal.

Response: The experiments were done several years ago. Thus, is meanwhile not possible to show the viability of cells during two, five and nine days of ethanol treatment, and 4 day withdrawal because the cell line no longer exists.

10)   If medium with ethanol was added every 48h it means that ethanol concentrations were not constant, due to high alcohol evaporation. It is possible that these cells in culture experienced many small ethanol withdrawals. The authors should have incubated the cells in alcohol-saturated atmosphere and changed the medium with alcohol every day. This is also a disadvantage of the study.

      Response: Thank you for this important comment. We have added this comment in the discussion as limitation.

11)   The authors wrote: “The (-1843/-1533) region contains 30 CpG sites; 4 of them could be analysed in the present study by pyrosequencing and commercially available CpG assays.” Please explain why the authors have chosen particularly these 4 CpG islands.

      Response: These 4 CpG sites were chosen because the availability of commercial CpG assays. For the other sites there are no assays available. This is now explained in the method section.

12)   The authors wrote: „The output data (obtained from Pyromark Q24 sequencing reactions), representing percent methylated cytosines over each of these three CpGs, was averaged for each sample analyzed.“  3 or 4 CpG island were analyzed? Please check.

Response: Four CpG sites were analyzed.

13)   Statistical analysis: The authors wrote. “Since all variables yielded significant deviation from normal distribution (Kolmogorov-Smirnov tests), non-parametrical tests (Mann-Whitney-U Test or Kruskal-Wallis analysis of variance) was employed for subsequent analyses.” However, age was compared between AD subjects and controls using Student’s t statistics. If there is not normal distribution in age the authors should use Mann-Whitney-U Test. Moreover, the authors should then use non-parametric Spearman’s test for relationships between age and CpG island methylation level, as well as Mann-Whitney-U Test for testing the influence and age.

Response: For the comparison of age between patients and controls, Mann-Whitney U Test was used. First paragraph of results section has been changed accordingly (Test-Value 675, p = 0.007). For the relationships between age and CpG island methylation levels and the influence of ages, Spearman’s rho and Mann-Whitney-U tests for the influence of age are used. No statistically significant results were obtained.

 14)   The authors wrote that they used two-way ANOVA and MANOVA repeated measures. However, the results only of ANOVA repeated measures were mentioned. Please check.

Response: ANOVA repeated measurement was employed only. Statistics section has been changed accordingly.

15)   The authors wrote: “Correction for multiple testing (Bonferroni) was applied when necessary”. However, there is no evidence of any correction for multiple testing. Please describe in what why the correction for multiple testing is applied and what p-value was considered significant after correction (since 4 CpG islands were analysed- p value should be set at least at 0,0125).

Response: Thank you for the comment. Level of significance is set for p < 0.0125. Statistics section has been changed accordingly.

Results

  • The authors wrote. „No significant difference between alcohol-dependent individuals and healthy controls were found for age (t-value 2,83, df = 87, p = 0,006).“ However, AD patients were statistically significantly older then control subjects (p=0.006). Please check this.

Response: There is a statistically significant age difference between patients and controls. Results section has been changed accordingly.

  • Please present in Table the results regarding correlations between CpG methylation sites and age in alcohol-dependent patients and controls.

Response: Table 2 has been added presenting the correlations between CpG methylation and age at baseline in alcohol-dependent individuals and controls.

  • Please present in Table the results regarding the influence of smoking status (smokers vs. non-smokers) on CpG site methylation.

Response: influence of smoking status on CpG site methylation is presented in the new table 3

  • There is no data presenting the percent methylated cytosines over each of these 4 CpGs. Please present the results (Figure) regarding the GABRA2 methylation in AD patients (admission) and controls, as well as regarding comparison of GABRA2 CpG Methylation sites over time in AD subjects.

Response: figure 2 has been added to present results on CpG site methylation in AD subjects and controls at baseline and follow-up (AD patients only).

  • Please present all data regarding influence of various alcohol dependence characteristics on GABRA2 CpG Methylation sites over time in AD subjects.

Response: Table 4 has been added to present all alcohol-related characteristics and their influence on CpG methylation.

  • All data should be presented in forms of Tables and Figures and with statistical tests used specified for each analysis.

Response: tables 2 to 4 and figure 2 have been added, including relevant statistics.

  • Please present the % of Methylation at four CpG sites of the GABRA2 gene in all groups of neuroblastoma cells (control, ethanol treated, withdrawal) as Figure.

Response: The investigated four CpG-sites were not methylated in neuroblastoma cells. Thus, we can not show the results in a figure. We mentioned this now in the result section.

          In addition, present the viability of neuroblastoma cells during ethanol treatment, since these high concentrations of ethanol are very toxic to cells.

Response: The experiments were done several years ago. Thus, is meanwhile not possible to show the viability of cells during two, five and nine days of ethanol treatment, and 4 day withdrawal because the cell line no longer exists.

      Discussion

Please add other limitations of the study (no female subjects, not controlling for early and late onset of alcohol abuse, too high toxic ethanol concentrations used for neuroblastoma treatment, etc.) and discuss them.

Response: limitation to the study have been added in a paragraph at the end of the discussion section. 

Minor

Common abbreviation for standard deviation is SD

Response: abbreviation has been corrected. Thank you for noting.

There are some typos and language errors throughout the text, which need to be corrected.

Response: the text has been re-read and edited carefully.

Reviewer 2 Report

This manuscript presents new data concerning the potential role of alcohol exposure on DNA methylation of GABRA2 gene.  In view of the numerous studies showing a possible relationship between GABRA2 and the development of alcohol use disorders (AUDs), this study offers relevant information regarding DNA methylation as a possible mechanism in the development of AUD.

There are several positive features of the study that was undertaken.  These include a careful assessment of alcohol dependence using a structured validated instrument (SSAGA), and enrollment of persons coming to treatment shortly after entering treatment and again following recovery.  Study goals were enriched by completion of an experiment using neuroblastoma cells exposed to alcohol.  With the small sample size, it is difficult to demonstrate an absence of an effect of chronic alcohol use on GABRA2  DNA methylation.  However, the authors have performed power analyses and presented these showing adequate power to have seen a result if one were present.

There are areas of the manuscript that would benefit from clarification by the authors.

(1) Section 2.2 - Control Group: Although it is stated that this group was free of mental and physical disorders, no mention is made of the instrument used to establish this.  If the SSAGA was used this should be stated as it lends confidence that the controls were evaluated as extensively as were the alcohol dependent group.

(2) Commercially available Pyromark assays are typically offered with control samples for comparison with the user's experimental samples.  If these were run, the results should be included.  Also, was the choice of the region for analysis based on the assessment described in Section 2.4 or rather the availability of a commercially available assay.  Also, when describing "older bisulfite sequencing data" that led to this choice, it was unclear if this was results from previous work with the same Pyromark assay or from an independent analysis that included more CpG sites.

(3) The average value across the 3 CpG sites was used for the analysis.  This is appropriate and is usually suggested by Pyromark.  The departure from normality of these mean values was analyzed using non-parametric statistical analyses.  This is appropriate, however a parametric analysis might be more powerful and might reveal significant differences.  One approach to achieving this would be to use a log transform of the mean methylation data.  Because the results show no significant differences, it would be useful to offer this analysis as well.

(4) The authors have attempted to put there negative results in the context of previous published studies showing no relationship between GABRA2 and alcohol use disorders by citing references 19, 20, and 21.  I would suggest adding Matthews et al (2007).  While reference 20 shows a lack of relationship in a large community sample of over 7,000 individuals, these participants would not be expected to be a severely affected based on family history of multigenerational, multiplex presence of AUD.  One of the original positive findings was from the COGA study based on multiplex families (9).   The Matthews et al (2007) paper used multiplex families providing a negative result for participants with presumably more intensive genetic loading for AUDs.

Author Response

Manuscript medicina-1940573

Responses to Review 2:

This manuscript presents new data concerning the potential role of alcohol exposure on DNA methylation of GABRA2 gene.  In view of the numerous studies showing a possible relationship between GABRA2 and the development of alcohol use disorders (AUDs), this study offers relevant information regarding DNA methylation as a possible mechanism in the development of AUD.

There are several positive features of the study that was undertaken.  These include a careful assessment of alcohol dependence using a structured validated instrument (SSAGA), and enrollment of persons coming to treatment shortly after entering treatment and again following recovery.  Study goals were enriched by completion of an experiment using neuroblastoma cells exposed to alcohol.  With the small sample size, it is difficult to demonstrate an absence of an effect of chronic alcohol use on GABRA2  DNA methylation.  However, the authors have performed power analyses and presented these showing adequate power to have seen a result if one were present.

There are areas of the manuscript that would benefit from clarification by the authors.

(1) Section 2.2 - Control Group: Although it is stated that this group was free of mental and physical disorders, no mention is made of the instrument used to establish this.  If the SSAGA was used this should be stated as it lends confidence that the controls were evaluated as extensively as were the alcohol dependent group.

Response: The description of the control sample was expanded by the following explanation: “All controls were assessed comprehensively to exclude medical and mental diseases including Axis I/II disorders, such as schizophrenia, depression, personality disorders or any alcohol- and substance use disorders by a short structured interview with an experienced psychiatrist, as well as with the personality questionnaires (MMPI, NEO-FFI, TCI) and a routine laboratory screening”.

(2) Commercially available Pyromark assays are typically offered with control samples for comparison with the user's experimental samples.  If these were run, the results should be included.  Also, was the choice of the region for analysis based on the assessment described in Section 2.4 or rather the availability of a commercially available assay.  Also, when describing "older bisulfite sequencing data" that led to this choice, it was unclear if this was results from previous work with the same Pyromark assay or from an independent analysis that included more CpG sites.

Response:  The used pyromark assays did not include any controls. Thus, we can not provide the mentioned control data.

Within some older bisulphite sequencing approaches by capillary electrophoresis of our group, we found intensive methylation from nucleotide -1843 to -1533. In the other sequenced parts of the CpG island there was no, respectively very rare, negligible methylation. Thus, we concentrated to the methylated region from nucleotide -1843 to -1533. The (-1843/-1533) region contains 30 CpG sites; 4 of them could be analysed in the present study by pyrosequencing and commercially available CpG assays. These 4 CpG sites were chosen because the availability of commercial CpG assays. For the other sites there were no assays available. This information is now added in the method section. (3) The average value across the 3 CpG sites was used for the analysis.  This is appropriate and is usually suggested by Pyromark.  The departure from normality of these mean values was analyzed using non-parametric statistical analyses.  This is appropriate, however a parametric analysis might be more powerful and might reveal significant differences.  One approach to achieving this would be to use a log transform of the mean methylation data.  Because the results show no significant differences, it would be useful to offer this analysis as well.

(3) The average value across the 3 CpG sites was used for the analysis.  This is appropriate and is usually suggested by Pyromark.  The departure from normality of these mean values was analyzed using non-parametric statistical analyses.  This is appropriate, however a parametric analysis might be more powerful and might reveal significant differences.  One approach to achieving this would be to use a log transform of the mean methylation data.  Because the results show no significant differences, it would be useful to offer this analysis as well.

Response: The CpG site methylation variables were log transformed and compared over the 3 measures using ANOVA repeated measures. Again, no significant results were obtained (CpG sites 1-3 F-value: 0.266, p = 0.768. 

(4) The authors have attempted to put there negative results in the context of previous published studies showing no relationship between GABRA2 and alcohol use disorders by citing references 19, 20, and 21.  I would suggest adding Matthews et al (2007).  While reference 20 shows a lack of relationship in a large community sample of over 7,000 individuals, these participants would not be expected to be a severely affected based on family history of multigenerational, multiplex presence of AUD.  One of the original positive findings was from the COGA study based on multiplex families (9).   The Matthews et al (2007) paper used multiplex families providing a negative result for participants with presumably more intensive genetic loading for AUDs.

Response: the publication of Matthews et al has been added to the introduction section

Round 2

Reviewer 1 Report

The authors have accepted some but not all reviewers suggestions. The manuscript is improved; however there are still some issues that need to be corrected. The manuscript is poorly written and there are still some typos and language errors throughout the text, which need to be corrected.

Introduction

The authors still wrote that neuroblastoma cell cultures were exposed to several dosages of alcohol and underwent subsequent alcohol withdrawal. This should be corrected.

Materials and methods

The authors still wrote: After 4-day incubation, EtOH (200 mM) was added every 48h during medium replenishment, if necessary. This should be corrected.

In the text the authors should add the explanation why have you used 100 mM ethanol in cell culture experiments. They should also try to compare the usual concentration of ethanol in the blood and brain of alcohol-dependent patients with the ethanol concentration that they have used in the cell culture.

Statistical analysis: The authors still write that they used two-way ANOVA instead of ANOVA repeated measures. This should be corrected.

Results

Table 1 lacks statistical analysis of daily alcohol intake and smoking frequency between control and AD patients. Present these findings in the paper and discuss the age and smoking differences in the section discussion.

Authors wrote: „Within the alcohol-dependent patients and controls, no significant correlations between CpG methylation sites and age were found. For subsequent analyses within the alcohol-dependent group, age was not used as a co-factor.“

However, in the Table 2 the significant correlation (p=0.005) between CpG island 4 basline methylation and age is presented. This should be corrected and discussed in the Discussion.

Tables 3 and 4: Please present all exact p-values instead only writing n.s.

In the results the authors should present the data obtained in the analysis using subgrouping the individuals in “early vs. late” onset of alcohol abuse (statistics and all numbers as a Table). This should be also discussed in the Discussion.

The authors should show the viability of cells during two, five and nine days of ethanol treatment, and 4 day withdrawal. Neuroblastoma cells are common, commercially available cell line so there should be no problem to treat the cells again for 2 weeks with 100 mM ethanol in order to check for their viability during ethanol treatment and withdrawal.

Author Response

Responses to Reviewer 1:

The authors have accepted some but not all reviewers suggestions. The manuscript is improved; however there are still some issues that need to be corrected. The manuscript is poorly written and there are still some typos and language errors throughout the text, which need to be corrected.

Response: The manuscript is cross-read by a native speaker and corrected.

Introduction

The authors still wrote that neuroblastoma cell cultures were exposed to several dosages of alcohol and underwent subsequent alcohol withdrawal. This should be corrected.

Response: the sentence at the end of the introduction section has been corrected. Thanks for noting.

Materials and methods

The authors still wrote: After 4-day incubation, EtOH (200 mM) was added every 48h during medium replenishment, if necessary. This should be corrected.

Response: Corrected. Thanks for noting.

In the text the authors should add the explanation why have you used 100 mM ethanol in cell culture experiments. They should also try to compare the usual concentration of ethanol in the blood and brain of alcohol-dependent patients with the ethanol concentration that they have used in the cell culture.

Response:  Currently, we do not have data on brain vs. blood concentrations in patients. For e.g. cerebrospinal fluid alcohol concentration, lumbar puncture of patient would be needed which is not part of the study plan and requires a separate informed consent and a different study plan.

Statistical analysis: The authors still write that they used two-way ANOVA instead of ANOVA repeated measures. This should be corrected.

Response: The sentence in statistical analysis has been changed. Thanks for noting.

Results

Table 1 lacks statistical analysis of daily alcohol intake and smoking frequency between control and AD patients. Present these findings in the paper and discuss the age and smoking differences in the section discussion.

Response: p-value have been included into table 1. Age and smoking differences are included into the discussion section. 

Authors wrote: „Within the alcohol-dependent patients and controls, no significant correlations between CpG methylation sites and age were found. For subsequent analyses within the alcohol-dependent group, age was not used as a co-factor.“

Response: the sentence was corrected, and age is used as a co-factor in ANOVA. Correspondingly, results in table 4 have been adapted.   

However, in the Table 2 the significant correlation (p=0.005) between CpG island 4 baseline methylation and age is presented. This should be corrected and discussed in the Discussion.

Response: Results have been corrected. Age is used as a co-factor. Influence of age is mentioned in the discussion section.

Tables 3 and 4: Please present all exact p-values instead only writing n.s.

Response:  Tables 3 and 4 have been corrected, exact p – values are now included.

In the results the authors should present the data obtained in the analysis using subgrouping the individuals in “early vs. late” onset of alcohol abuse (statistics and all numbers as a Table). This should be also discussed in the Discussion.

Response: Results on early vs. late-onset patients are now presented in table 5. Together with age and smoking, the negative relationship between early onset and methylation is included into the discussion section. 

The authors should show the viability of cells during two, five and nine days of ethanol treatment, and 4-day withdrawal. Neuroblastoma cells are common, commercially available cell line so there should be no problem to treat the cells again for 2 weeks with 100 mM ethanol in order to check for their viability during ethanol treatment and withdrawal.

Response: The cell culture tests presented in this work were carried out several years ago. In the meantime, the laboratory structures and responsibilities have changed in such a way that we unfortunately no longer have the opportunity to carry out further cell culture experiments. We agree with the reviewer's comments but regret that further investigation is no longer possible.

Round 3

Reviewer 1 Report

The authors have accepted some but not all reviewers suggestions. The manuscript is improved; however there are still some issues that need to be corrected. The authors wrote that they have corrected some issues but actually they did not. 

Introduction: The authors still wrote that neuroblastoma cell cultures were exposed to several dosages of alcohol and underwent subsequent alcohol withdrawal (page 5, line 110). The authors wrote that they have corrected it but they did not. It should be corrected. I am wasting my time by correcting the same mistakes. 

In the text the authors should add the explanation why have you used 100 mM ethanol in cell culture experiments. It is known that people are very drunk when they have 0.2% alcohol in the blood (blood alcohol content, BAC). 0.2% corresponds concentration of 21.7 mM ethanol in the blood. The alcohol concentrations that reach the brain of alcohol-dependent people are probably much lower then in blood. The authors use 100 mM ethanol in cell culture.  This is high ethanol concentration which may be toxic to neuronal cells.

One of the major concerns with the in vitro alcohol treatment using concentrations above 100 mmol/L is the direct cytotoxic effect of alcohol on cells [Dolganiuc and Szabo 2009].  Therefore, to investigate whether 100 mM ethanol is toxic to neuroblastoma cells or not, viability of cells should be determined. If authors can not perform the experiments to asses the cell viability, they should in the discussion acknowledge as the limitation of the study the usage of high concentrations of alcohol which may be neurotoxic and the should acknowledge that viability of cells should be determined in future.

Statistical analysis: The authors still write that they used two-way ANOVA repeated measures. The authors wrote that they have corrected it but they did not. It should be corrected. 

In the Introduction and Discussion the authors should dicuss the rationale why they have performed the analysis using subgrouping the individuals in “early vs. late” onset of alcohol abuse. Early  onset of alcohol use corresponds to type II alcoholism subgroup according to the Cloninger's classification (Cloninger et al., 1988). Type II or “male-limited” alcoholism is suggested to be strongly heritable (estimated heritability of 90%), transmitted primarily from father to son, and showing moderate environmental influence (Cloninger et al., 1988). Moreover, early onset of alcoholism often reflects greater severity, including a higher risk for recurrence.

Author Response

Responses to Reviewer 1:

The authors have accepted some but not all reviewers’ suggestions. The manuscript is improved; however there are still some issues that need to be corrected. The authors wrote that they have corrected some issues but actually they did not. 

Introduction: The authors still wrote that neuroblastoma cell cultures were exposed to several dosages of alcohol and underwent subsequent alcohol withdrawal (page 5, line 110). The authors wrote that they have corrected it but they did not. It should be corrected. I am wasting my time by correcting the same mistakes. 

Response: Sorry, this was overlooked. Corrected.

In the text the authors should add the explanation why have you used 100 mM ethanol in cell culture experiments. It is known that people are very drunk when they have 0.2% alcohol in the blood (blood alcohol content, BAC). 0.2% corresponds concentration of 21.7 mM ethanol in the blood. The alcohol concentrations that reach the brain of alcohol-dependent people are probably much lower then in blood. The authors use 100 mM ethanol in cell culture.  This is high ethanol concentration which may be toxic to neuronal cells.

Reponse: We added the explanation in the method section and in the discussion.

One of the major concerns with the in vitro alcohol treatment using concentrations above 100 mmol/L is the direct cytotoxic effect of alcohol on cells [Dolganiuc and Szabo 2009].  Therefore, to investigate whether 100 mM ethanol is toxic to neuroblastoma cells or not, viability of cells should be determined. If authors can not perform the experiments to asses the cell viability, they should in the discussion acknowledge as the limitation of the study the usage of high concentrations of alcohol which may be neurotoxic and the should acknowledge that viability of cells should be determined in future.

Response: We added a paragraph in the discussion were we consider all the limiting factors and interpret the cell culture results very cautiously.

Statistical analysis: The authors still write that they used two-way ANOVA repeated measures. The authors wrote that they have corrected it but they did not. It should be corrected. 

Response: Thanks for noting. Corrected.

In the Introduction and Discussion the authors should dicuss the rationale why they have performed the analysis using subgrouping the individuals in “early vs. late” onset of alcohol abuse. Early  onset of alcohol use corresponds to type II alcoholism subgroup according to the Cloninger's classification (Cloninger et al., 1988). Type II or “male-limited” alcoholism is suggested to be strongly heritable (estimated heritability of 90%), transmitted primarily from father to son, and showing moderate environmental influence (Cloninger et al., 1988). Moreover, early onset of alcoholism often reflects greater severity, including a higher risk for recurrence.

Response: the relevant passage has been introduced into the introduction section and is further elaborated on in the discussion section.